# Centralization in Decentralized Web: Challenges and Opportunities in IPFS's Data Management

## Abstract

The InterPlanetary File System (IPFS) is a pioneering effort for Web 3.0, well-known for its decentralized infrastructure. However, some recent studies have shown that IPFS exhibits a high degree of centralization and has integrated centralized components for better performance. While this change contradicts the core decentralized ethos of IPFS and introduces risks of hurting the data replication level and thus availability, it also opens some opportunities for better data management and cost savings through deduplication.

To explore these challenges and opportunities, we start by collecting an extensive dataset of IPFS internal traffic spanning the last three years with 20+ billion messages. By analyzing this long-term trace, we obtain a more complete and accurate view of how the status of centralization evolves over an extended period. In particular, (1) IPFS shows a low replication level in general, with only about 2.71% of data files replicated more than 5 times. While increasing replication enhances lookup performance and data availability, it adversely affects downloading throughput due to the overhead involved in managing peer connections, (2) there is a clear growing trend in centralization within IPFS in the last 3 years, with just 5% of peers now hosting over 80% of the content, significantly decreasing from 21.38% 3 years ago, which is largely driven by the increase of cloud nodes, (3) the IPFS default deduplication strategy using Fixed-Size Chunking (FSC) is largely inefficient, especially with the current 256KB chunk size, achieving nearly zero efficiency. Although Content-Defined Chunking (CDC) with smaller chunks could save significant storage (about 1.8 PB) and cost, it could impact user performance negatively. We thus design and evaluate a new metadata format that optimizes deduplication without compromising performance.

**Relevance to WWW:** This study conducts a web measurement study utilizing a 3-year dataset from IPFS network.

## 1 Introduction

From the underlying infrastructure support to the way how users interact with the web, there has been a clear trend toward centralization in the current web along the wide adoption of cloud computing over the past two decades. Although this undeniably provides convenient services for managing the web data and serving the users, it also raises growing concerns about data control and monopolies by big techs, censorship and content moderation, monetization and fairness, among others.

In response to those concerns, recent years have witnessed a growing surge in the technology campaign referred to as "Web 3.0" or "Decentralized Web", such that no individual entity can control or censor the entire operation of the system. One of the leading efforts in this movement is the InterPlanetary File System (IPFS) [7]. IPFS uses a Peer-to-Peer (P2P) architecture, where identical data copies are distributed and shared among multiple peers in the system. Upon a request, a client can be served by any of available peers having a copy of the data. The adoption of IPFS is widespread, with over 250K active daily nodes and spanning 152 countries [27]. It also serves as the storage layer for various Decentralized Applications, including Non-Fungible Token (NFT) [30].

Although IPFS is meant to be a decentralized system, some recent studies [6, 22] have looked into snapshots of the client accesses in IPFS and shown that IPFS exhibits a surprising degree of centralization when serving the client requests, where a small group of cloud-based nodes serve the majority of the client accesses. The centralization in IPFS has indeed improved performance (as users are often served by faster cloud nodes) and enabled easier client access by integrating gateways to serve users not running the IPFS protocol directly [31]. However, such centralization has many downsides as mentioned before and raises questions on whether this is due to a limited number of copies of each data file distributed in the system, and more importantly, whether such observations are temporal effects due to dynamic system factors, such as peer churns, and thus do not fully capture the complete state of IPFS.

On the other hand, regardless of the underlying reasons, from a practical standpoint, such centralization, if it exists, could present an opportunity for improved web data management and cost savings through deduplication [33], which is important for those cloud nodes. By default, IPFS employs deduplication locally to ensure faster publication performance and uses a fixed-size chunking (FSC) [19] for data deduplication. However, how efficient such data deduplication in IPFS remains unclear.

To explore answers to the above questions, we began our study by massively collecting the IPFS internal content exchange traces in the past 3 years, spanning from March 2021 to August 2024, with over 20 billion messages. This 3-year trace of internal IPFS traffic can help reveal the evolving of

centralization changes, given the instant snapshots used in previous analysis [22]. By performing a systematic measurement and analysis on these traces, we aim to answer the following key questions: **RQ1:** how are the identical data copies distributed? i.e., what is the data replication level in current IPFS and how does it impact the users' accesses? **RQ2:** how has the decentralization of IPFS evolved over the past three years and what is the trend of the changes? Our measurement and analysis lead to the following findings.

- **[Replication]** IPFS exhibits a low replication level with only about 29.20% of CIDs replicated more than once, and a mere 2.71% more than 5 times. Increasing replication in IPFS generally enhances lookup performance due to a higher possibility of being discovered. However, the increased replication negatively impacts downloading throughput after a certain level as a result of the extra overhead involved in consistently choosing (and switching to) better peers for downloading.
- **[Centralization]** There is a significantly growing trend of centralization in IPFS over the past 3 years, where 5% of the peers are now responsible for hosting 80.55% of the content in terms of storage capacity. In contrast, at the beginning of our measurement period, a more distributed 21.38% of the peers hosted 80% of the content. This increasing centralization can be attributed to the growing adoption of cloud nodes, whose share has increased from 50.02% to 87.33% over the same period.

The centralization trend further leads us to seek an answer to **RQ3:** how efficient is the default deduplication method (FSC) in IPFS? Our study shows the following.

- **[Deduplication]** Currently, IPFS achieves nearly zero deduplication efficiency by using the default FSC method with the default 256KB chunk size, where the storage savings could be upto 1.8 PB using Content-Defined Chunking method with a smaller chunk size such as 4KB. We further show that a smaller chunk size could negatively impact user performance in terms of the deduplication speed and downloading throughput. To balance the user performance and deduplication efficiency, we propose a new meatadata format exploiting storage locality to significantly reduce the IO overhead and improve the user performance comparable to the original IPFS.

Note that since IPFS is a distributed system, no one can collect all the exchange messages in this system. Ours is not an exception. However, by aggregating data over a 3-year period, we hope our data can more accurately capture the system image than previous studies. We hope our findings, presenting both challenges for decentralization and opportunities under centralization, can help Web 3.0 with its design and implementation. We will make the artifacts, including analysis scripts and properly anonymized data, publicly available upon the acceptance of this paper.

The remainder of the paper is as follows. A brief background of IPFS is presented in Section 2. We present our methodology of data collection in Section 3, measurement analysis regarding replication in Section 4, regarding centralization in Section 5 and regarding deduplication in Section 6. We discuss the related work in Section 7 and make concluding remarks in Section 8.

## 2 Background

**IPFS Overview.** IPFS is a decentralized Web 3.0 system that builds on top of a P2P network and facilitates a set of protocols. It uses content-based addressing, where each file is split into multiple chunks and each chunk is represented by an immutable, hashed and self-certifying Content Identifier (CID). The physical addresses of CIDs are stored in the Distributed Hash Table (DHT) [18]. This enables a peer, identified by a unique PeerID, to query the DHT to locate and retrieve the content without relying on a centralized server. For users not participating in the IPFS network, Gateway services are provided for them to access IPFS content via HTTP.

**DHT.** DHT is a key component of the IPFS routing system, responsible for storing and retrieving content. It indexes two types of records: provider records, which map CIDs to the nodes that advertise and provide the content, and peer records, which map PeerIDs to the physical addresses of nodes (*e.g.,* IP addresses). The DHT allows the user to locate which node is serving the target content and advertise its own content without the need for a centralized server.

**Content Publication and Deduplication.** IPFS operates at the block level as each file is first split into multiple chunks. These chunks form a Merkle DAG, where each node corresponds to a block. Specifically, raw data is stored in leaf blocks, while parent blocks contain metadata indicating how the original file is segmented and can be reconstructed. Similar to centralized deduplication systems, IPFS achieves deduplication at the chunking stage by ensuring that no identical chunks are stored. The default deduplication algorithm in IPFS is Fixed Size Chunking (FSC), set at a default size of 256 KB. During the process of chunking, the CID of each chunk is computed based on its content. The CID can be configured into two interchangeable formats: version0 and version1 [3]. Additionally, IPFS supports configurable algorithms like Rabin Fingerprint [8] and Buzhash [9]. Unlike centralized storage systems, which typically perform deduplication processes on a central server and distribute the blocks afterwards, IPFS performs deduplication locally.

**Content Retrieval and Bitswap** Content retrieval in IPFS involves two main stages: lookup and downloading. In the lookup stage, IPFS initially attempts to find which peers have the requested content. Before querying the DHT, IPFS uses

the Bitswap [1], the file exchange protocol in IPFS, to send a WANT-HAVE message to nearby peers, asking if they have the desired content. If no peers respond within a 1-second threshold, IPFS resorts to querying the DHT to locate the content. Once the content is located, the process enters the downloading stage, where Bitswap requests the actual data by sending a WANT-BLOCK message to peers holding the required blocks. Bitswap then transfers the actual data blocks, completing the retrieval process.

## 3 Methodology

### 3.1 Data Collection

**Bitswap Logs.** We collect Bitswap traces from March 1st, 2021 to Aug 15th, 2024 using a modified IPFS node implementation with unlimited connections with peers (details similar to [5]). Our crawler collects all incoming 1-hop Bitswap broadcast traffic to disk. We log the timestamp, the sender's PeerID and network address, the type of request, the receiver's PeerID and the target CID. From this, we observe approximately 21 M requests daily on average with a total of 1.8 B unique CIDs.

**DHT Logs.** We also set up a modified version of a DHT server to collect the IPFS DHT traffic. We set up 2 virtual peer IDs and log all the incoming DHT requests to disks. The content of the DHT request is similar to the Bitswap messages, containing the sender's PeerID and network address, the type of request, and the target PeerID or CID denpending of the request type. The collection of DHT traffic was also conducted from March 1st, 2021 to Aug 15th, 2024 From this, we observe about 1 M requests daily on average, covering 120 M unique CIDs.

### 3.2 Ethical Consideration

This work is conducted under an IRB approval from our institution. The details of our ethical considerations are in Appendix A.

## 4 Data Replication

In this section, we start by examining the degree of replication within IPFS. Then we look at how different degrees of file replication across the network influence the efficiency and reliability of retrieving the content.

### 4.1 Degree of Replication

**Methodology.** Using the DHT logs and Bitswap logs, we have constructed a CID-provider mapping that retains only the CIDs representing a complete file or directory. This filtering process is facilitated by identifying the request pattern for CIDs in the logs, where a CID that corresponds to a complete file or directory is typically requested first. This

allows us to bypass the need to download the CID to verify its contents.

Based on the CID versioning of IPFS, we further categorize the CIDs into two classes: version0 and version1. It is important to note that while different CID versions of a file can be inter-converted, they represent distinct objects as IPFS solely rely on CID to locate a file.

To verify replication accuracy, we send a WANT-HAVE message to the peer who is assumed to be the provider. If the recipient fails to respond, we repeat this query every 4 hours for one week. If there is still no response after this period, we consider that the CID no longer belongs to the peer. This method ensures that we are not only mapping CIDs to providers correctly but also verifying the presence and availability of the content they claim to host.

**Result and Analysis.** Figure 1 displays the cumulative distribution of CIDs categorized by version0 and version1. In this study, we detect a total of 214 million CIDs, with 147 million belonging to version0 and 67 million to version1. The red curve in the figure represents the cumulative distribution when all CIDs are converted to a single version, illustrating the overall trend.

The data reveals a low level of replication across the network. Specifically, only 29.20% of CIDs are replicated more than once, and a mere 2.71% of CIDs experience replication more than five times. This indicates that while the network is capable of hosting multiple copies of the same content, the actual practice of replication is not extensively utilized.

Interestingly, the analysis uncovers a significant amount of "replication wastage". We find that 18.24 million files have both version0 and version1 CIDs, which artificially inflates the count of unique content due to version differences.

> **Takeaway 1:** While IPFS is designed as a file sharing system, its file replication level typically remains low with only about 30% of its files replicated more than once. Moreover, the issue is exacerbated by the fact that different CID versions can be generated for the same file, which inflates the perceived uniqueness of files within the system.

### 4.2 Replication Impact on Performance

**Methodology.** To study how the replication level impacts the file retrieval performance in IPFS, we first set up an IPFS client (a t2.medium EC2 instance with 2 vCPUs and 4GB of memory) in central Europe. We let this IPFS client fetch 1,000 files, 10 MB each, in each replication level from 1 to 20. We timestamp the time to discover the provider (lookup time) and the time to fetch the blocks (downloading time). We then calculate the throughput by dividing the downloading time instead of the entire retrieval time. It is also important to

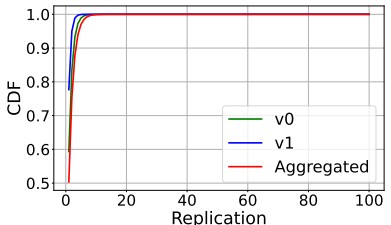

Figure 1: CID replication level in version 0, version 1 and after aggregation.

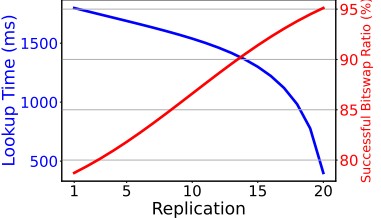

Figure 2: The performance of lookup in different replication levels.

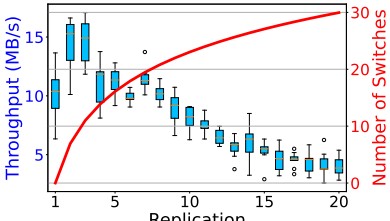

Figure 3: The performance of throughput in different replication levels.

note that the provider can be discovered by Bitswap before launching the DHT as illustrated in Section 2. As such, we further examine their Bitswap success ratio as a function of replication level.

**Results and Analysis.** Figure 2 demonstrates the average lookup time (on the left *y-axis*) and the Bitswap success ratio (on the right *y-axis*) at each replication level. As can be seen in this figure, the replication level can significantly improve the lookup time performance. Specifically, the average lookup time decreases from 1817 ms at a replication level of 1 to 397 ms at a replication level of 20, while the Bitswap success ratio concurrently rises from 73.21% to 95.09%. This improvement is intuitive; as content is held by more peers, it becomes more readily detectable by Bitswap and DHT mechanisms, thereby reducing the time required for lookups.

Figure 3 shows the downloading throughput (on the left *y-axis*) at different replication levels. Different from the consistently improving trend observed in lookup time as the replication level increases, the downloading throughput initially improves but then decreases. Specifically, the average throughput peaks at 14.54 MB/s at the replication level of 2. Beyond this point, the throughput consistently decreases as the replication level increases.

To understand the underlying dynamics of this pattern, we further look into its downloading phase driven by Bitswap. We notice that there is a clear increasing trend of the request-peer switch as depicted in the blue curve in Figure 3. This 'switch' refers to the action of changing the downloading peer to another peer during the file retrieval process. The switching is to get a better response time and balance the specific traffic from certain provider. However, while switching can optimize the download process under certain conditions, excessive switching can be detrimental. It involves consistently closing and reopening connections, as well as re-requesting the target block from a new source. This added complexity and overhead can significantly delay the entire downloading process, as each switch consumes time and potentially disrupts the steady flow of data transfer.

> **Takeaway 2:** While increasing the replication level in IPFS improves lookup performance due to a higher Bitswap success ratio, it negatively impact the downloading throughput after a certain level because of the extra overhead involved in consistently choosing (and switching to) better peers to download from.

## 5 Measurement of Centralization

Given the low replication level in IPFS found in the last section, we wonder how the content distribution changed within our 3-year dataset - potentially centralization as reported by [6, 22]. To this end, in this section, we aim to analyze the evolution of IPFS's centralization over a 3-year period by employing statistical measures such as the Gini Coefficient and Shannon entropy to quantify its centralization level.

### 5.1 Methodology

**Entropy.** Entropy quantifies the unpredictability or randomness in the output of an information source. For IPFS, the distribution and frequency of file access define the probability distribution. Files that are accessed more frequently demonstrate lower entropy, indicating reduced uncertainty and potential centralization in file access patterns. To quantify the centralization level as entropy, we have

$$H = -\sum_{i=1}^{n} p_i \log_2 p_i$$

where $n$ is the number of unique files/CIDs accessed while $p_i = \frac{\text{Accesses to file } i}{\text{Total accesses to all files}}$.

**Gini Coefficient.** The Gini coefficient is another useful metric, commonly utilized to measure inequality in distributions [29]. In the context of IPFS, it is used to quantify the inequality of storage distribution. The formula to calculate the Gini coefficient is as follows:

$$G = \frac{\sum_{i=1}^{n} \sum_{j=1}^{n} |x_i - x_j|}{2n^2 \overline{x}},$$

where $n$ is the number of nodes, $x_i$ is the amount of data stored by node $i$, and $\overline{x}$ is the mean amount of data stored per node.

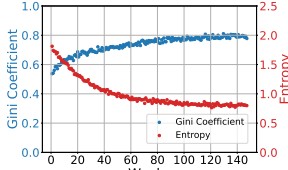 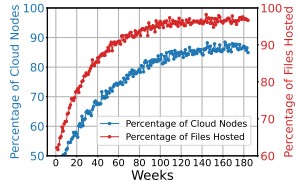

**Figure 4: Gini Coefficient and entropy over the 3-year time period.**

**Figure 5: The share of cloud nodes and their host files over the 3-year time period.**

**Data Synthesis.** Using the Biswap and DHT logs, we analyze changes in the centralization level over time by examining data weekly. Each week, we calculate the number of times a CID is requested and identify which peers own these CIDs. Table 1 lists the characteristics of this dataset. For this analysis, we make three key assumptions: (1) each CID represents a storage block of 256 KB, as this is the default block size of IPFS, (2) the peers hosting these CIDs remain consistently active in the network and continue to provide access to the CIDs without leaving the network, and (3) for those peers announcing multiple IP addresses as a result of peer churn, we count them as a single entity, *i.e.,* provider.

**Table 1: Dataset characteristics used in the centralization level analysis.**

| CID | Requested By | Owned BY | Timestamp |
| --- | --- | --- | --- |
| $c_0$ | $<...p_i...>$ | $<...p_j...>$ | $Week_k$ |

## 5.2 Results

Figure 4 depicts the changes in the Gini coefficient and entropy over the three-year period. Initially, in early 2021, the Gini Coefficient starts at 0.53, indicating a moderate level of storage inequality among peers. Over the course of this period, this coefficient increases significantly, reaching 0.78 by mid-2024. This rise signifies that storage inequality has become increasingly severe, with a small number of powerful peers accumulating a larger share of content. Specifically, by the last week of our measurement period, only 5% of the peers are responsible for hosting 80.55% of the content. In contrast, at the beginning of our measurement period, 21.38% of the peers hosted 80% of the content.

Similarly, the entropy of IPFS follows a comparable pattern to the Gini Coefficient, exhibiting a decreasing trend over the three years. The entropy metric measures the predictability of file access within the network; a decline in entropy suggests that access to popular contents has become more frequent and predictable. This decrease in entropy, along with the rise in the Gini Coefficient, points to a growing level of centralization in IPFS, highlighting a shift towards more centralized control within the IPFS network.

To understand why such centralization trends may be occurring, we are inspired by prior studies [6, 22], which

suggest that the rise of centralized cloud nodes could be the reason. By identifying the IP addresses, we can distinguish cloud peers located in data centers and gateway nodes maintained for users without direct access to IPFS. Obtaining the IP addresses and PeerID of these gateway nodes is straightforward as they typically do not leave the network, making them easy to track.

We further analyze the share of storage these cloud nodes manage. Figure 5 depicts the percentage change of cloud nodes and the files they serve over the 3-year period. The data shows a clear increasing trend. For instance, at the start of our study period, cloud nodes comprised 50.02% of the peer set and hosted 52.32% of the files. By the end of the period, these figures dramatically increased to 87.33% of the peer set and 97.43% of the total files. This indicates an increasingly dominant role of cloud nodes within the IPFS network.

Our findings suggest a higher percentage of cloud node involvement than the results from previous studies [6], which estimated the percentage by crawling the DHT and building a network topology. Our results are derived directly from internal IPFS traffic data, which explains the greater share attributed to cloud nodes. More importantly, we observed a concerning trend towards centralization within our measurement period. This increased reliance on cloud nodes suggests a movement towards centralization within an ecosystem that fundamentally aims for decentralization.

> **Takeaway 3:** Both the entropy and Gini Coefficient studies clearly show a significant centralization trend in IPFS over the 3-year dataset, suggesting that in practice, IPFS is on the opposite direction of its decentralized goal. As a result, file access in IPFS becomes more predictable and concentrated among fewer and powerful entities, further highlighting the centralization within IPFS.

## 6 Deduplication

In the last section, we show that the centralization has kept growing quickly in the past three years in IPFS. While this is on the opposite direction of its original design goal and deserves more research, from a practical standpoint view, serving the majority users from a small number of cloud based nodes does improve the user experience and attract users. Furthermore, this may allow these cloud nodes to efficiently perform deduplication for storage savings. To assess this, we begin by examining the efficiency of the default deduplication algorithm employed by IPFS, and compare with other alternatives. We then show the trade-offs involved in incorporating those deduplication methods, and propose and evaluate new solutions designed to balance these trade-offs.

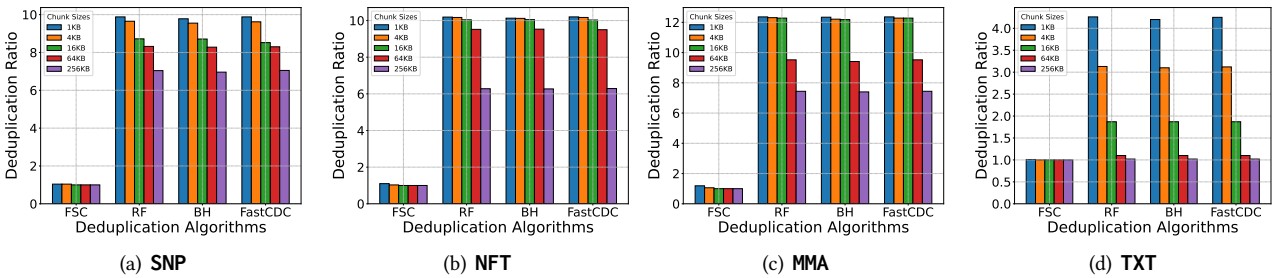

**Figure 6: Comparison of different deduplication algorithms on different datasets.**

## 6.1 Analysis of Deduplication Ratio

**Motivation.** Current IPFS employs a sequential and fixed-size chunking approach (FSC), where files are divided into multiple 256 KB chunks. However, FSC suffers from a boundary-shift problem that significantly lowers deduplication efficiency. For example, if one single bit is deleted at the beginning of a file, all current chunk cutpoints (i.e., boundaries) declared by FSC will be shifted and no duplicate chunks will be detected. As such, in order to evaluate the deduplication technique employed by IPFS, we perform our deduplication ratio measurements on four traces, detailed in Table 2. (1) SNP: This trace is the "snapshot" of top 180 providers by sampling 1% the CIDs in the Bitswap logs and DHT logs. (2) NFT: This is an NFT trace, scraped from the web of top-1000 NFT digital assets of OpenSea [2] that has an IPFS url. (3) MMA: This is a multimedia trace, obtained from a previous public gateway dataset [10]. (4) TXT: This is a text trace. This category includes plain text and JSON files from the same dataset as the multimedia traces, reflecting previous findings that these file types are among the most prevalent in IPFS.

**Table 2: Dataset characteristics used in the deduplication analysis.**

| Name | Size | # of Files | Description |
|------|------|-----------|-------------|
| SNP | 18.2 TB | 9.10 M | A snapshopt of the top-180 providers. |
| NFT | 6.7 TB | 8.22 M | Top-1000 collections of OpenSea. |
| MMA | 2.8 TB | 1.37 M | Multimedia files collected from a public gateway dataset [10]. |
| TXT | 13.3 GB | 4.01 M | Text and json files collected from a public gateway dataset [10]. |

**Methodology.** One common metric to evaluate deduplication efficiency is the deduplication ratio, which represents the ratio of the input dataset's size before and after deduplication. This metric reflects the ability of a deduplication technique to identify and eliminate duplicate data from the input.

For instance, a deduplication ratio of 2 indicates that 50% of the input data is redundant and can be eliminated. Another important metric is deduplication speed, which measures the volume of data processed within a given time frame. This metric assesses the efficiency of the deduplication process in terms of throughput.

To investigate the deduplication performance of the default FSC deduplication algorithm in IPFS, we utilize various chunk sizes ranging from 1KB to 1MB, incrementing each by powers of four. This approach allows us to understand how different chunk sizes impact the effectiveness of the FSC algorithm across the 4 datasets we employ. At the same time, we conduct controlled experiments to evaluate three alternative deduplication algorithms: Rabin Fingerprint (RF) [8], Buzhash (BH) [9], and FastCDC [34]. Each of these algorithms is configured to match the expected average chunk size used in the FSC approach. Our experiments are carried out using the original algorithms rather than the IPFS client to avoid other interferences. These tests are performed on an EC2 instance (t2.xlarge, 4 vCPUs, 16GB memory), applying each algorithm to the four datasets as specified.

**Results.** Figure 6 shows the deduplication ratio of different deduplication algorithms across different datasets. Notably, FSC shows a particularly low deduplication ratio, especially with the current default 256KB chunk size. For instance, FSC manages to eliminate only about 4% of duplicates in the snapshots of the top-180 providers. In stark contrast, content-based algorithms such as RF, BH, and FastCDC can reduce nearly 90% of duplicates in the same scenario, which can result in substantial storage savings of nearly 16 TB. These top-180 providers primarily use cloud services like AWS and Cloudflare. Given that our analysis sampled only 1% of the stored files, the total potential savings on these platforms could be around 1.8 PB. Considering the EC2 storage pricing of $0.08 per GB per month, the effective cost savings could exceed $100k per month. This significant financial impact highlights the economic benefits of optimizing deduplication strategies within IPFS, especially in cloud environments where storage costs are non-trivial.

**Takeaway 4:** The FSC achieves about zero deduplication efficiency at the default chunk size in all datasets. Applying CDC methods can save up to 90% storage. It is worthwhile especially considering that top providers in IPFS are mostly residing in the cloud centers.

## 6.2 Trade-off of Deduplication Algorithms

Although Content-Defined Chunking (CDC) based deduplication algorithm can effectively eliminate duplicates and save storage space, it is at the expense of high *chunking* overhead, the process of splitting file into chunks. To evaluate this overhead, we measure the deduplication speed in IPFS using a 100MB dummy file under different chunk sizes and deduplication algorithms. The measurement is performed on an EC2 instance (t2.xlarge). Note that we implement FastCDC in IPFS as it is not provided in the configurations and the chunking process only utilizes one thread. Table 3 presents the deduplication speed of different deduplication algorithms at different chunk sizes. As can be seen in this table, we observe that (1) FSC outperforms other 3 CDC based deduplication algorithms across all chunk sizes as it operates on a straightforward splitting of predetermined chunk size, (2) the deduplication speed becomes extremely slower when employing small chunk sizes like 1KB, despite these sizes offering better deduplication ratios.

**Table 3: Deduplication speed (MB/s) by algorithm and chunk size in original IPFS.**

| Alg. / Chunk Size | 1KB | 4KB | 16KB | 64KB | 256KB |
|:---:|:---:|:---:|:---:|:---:|:---:|
| FSC | 0.12 | 0.37 | 1.92 | 7.91 | 30.3 |
| RF | 0.08 | 0.22 | 0.92 | 3.45 | 15.25 |
| BH | 0.08 | 0.28 | 1.12 | 4.05 | 16.32 |
| FastCDC | 0.09 | 0.32 | 1.57 | 5.21 | 21.55 |

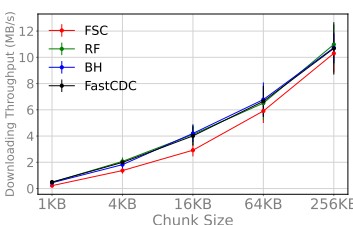

**Figure 7: The downloading speed (MB/s) as a function of chunk size.**

Another trade-off of applying the deduplication algorithm introduces is the downloading throughput. Previous works (*e.g.,* [22]) have shown that a small chunk size can be detrimental to the downloading throughput. Figure 7 further demonstrates this degradation with chunk size when downloading a 100MB file repeated 100 times. As shown in this figure, a 256KB chunk size offers a downloading throughput that is

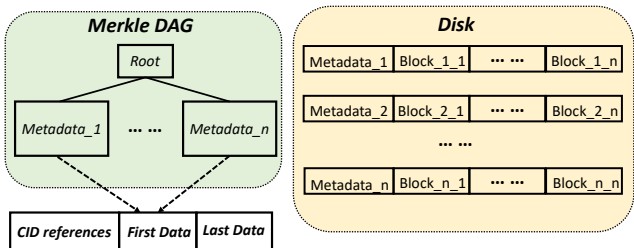

**Figure 8: Overview of the metadata design.**

50× larger than that of a 1KB chunk size when using the FastCDC. Although CDC can help improve the speed of data transfer as it can reduce the transferred data blocks, the improvement is minimal, improving by only 4.8% at a chunk size of 256KB compared to FSC. Nevertheless, the CDC method still experiences throughput degradation with decreasing chunk size. This significant difference underscores the need to balance the deduplication efficiency without compromising the overall performance of network data transfer and publication.

**Takeaway 5:** Typically, a smaller chunk size can facilitate a better deduplication ratio. However, such a small chunk size can negatively impact the user's performance: the deduplication speed and downloading throughput.

## 6.3 Metadata Based Deduplication

The relative poorer downloading and publication performance of IPFS is multidimensional not only because of the choice of the chunk size. There are other techniques like concurrent chunking or connections and employing dedicated nodes [31]to accelerate the performance of IPFS, which are orthoganal to the deduplication algorithms. Under the current framework of IPFS, we next discuss how to achieve higher deduplication ratio and maintain a comparable downloading and publication performance. As such, we propose a new metadata based deduplication technique for content delivery network like IPFS.

**Design of Metadata.** The poor performance of IPFS downloading stems from its linear processing manner using a small chunk size, linearly retrieving and writing small chunks. To improve the downloading performance and maintain a decent deduplication ratio, we introduce a new Merkle DAG as metadata for each IPFS file. The overview of the new Merkle DAG is presented in Figure 8. The core ideology is as follows. (1) *Metadata format.* In the original IPFS, the internal nodes of the Merkle tree merely contain the children CIDs. In contrast, the new metadata design incorporates additional details: it specifies the range of bytes that the metadata represents.

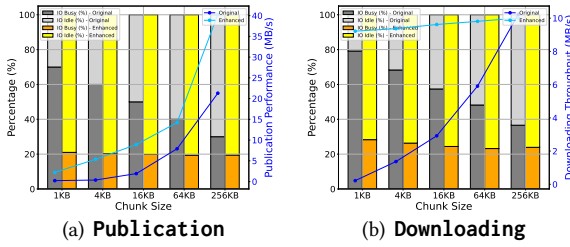

(a) **Publication**     (b) **Downloading**

**Figure 9: Performance comparison between the original IPFS and the enhanced IPFS.**

This is denoted as *"First Data"* and *"Last Data,"* indicating the relative positions of the metadata block. In addition, the new metadata references the CIDs based on their occurrence order and aggregates this with the count and size of occurrences in the form of *<CID, num, size>*, whereas the original IPFS system logs each CID sequentially without considering block locality. (2) *Storage locality.* Unlike the original IPFS, which stores blocks based on hash order to facilitate quick searches, our new design groups blocks that belong to the same file together and places the corresponding metadata adjacent to these blocks. For duplicated blocks, our system stores a pointer to the actual position. In other words, our new design organizes data file by file, as opposed to the original IPFS's method of storing data block by block.

In the original IPFS, files are fetched block by block. When using small chunk sizes, such as 1KB, the IPFS client must perform repetitive I/O operations to read and write these small blocks, which delays the entire downloading and publication process. Our proposed solution addresses this inefficiency by aggregating these small blocks into a larger block before any read/write operations on the disk, accelerating the downloading/publication process.

**Performance Evaluation.** In order to compare the enhanced IPFS with new metadata design, we deploy the enhanced IPFS on a private IPFS cluster with 5 nodes (EC2 instance t2.xlarge, HDD storage). We first let the original IPFS and enhanced IPFS upload a 1GB identical dummy file, respectively. We repeat the uploading 100 times and the IPFS client is re-configured after every experiment. In both systems, the duplicate data is removed by FastCDC algorithm and we log the disk time and the publication throughput. Figure 9(a) shows the result of the IO-busy percentage and the publication throughput. As shown in this figure, the average publication throughput is 2.24× larger than the original IPFS. This is due to the fact that the enhanced IPFS can largely reduce the I/O time as a result of sequential access of the disk, where the average I/O-busy percentage is 19.82% compared to 50.07% of the original IPFS. Then, we let the original IPFS and enhanced IPFS clients to fetch the uploaded 1GB file remotely and log the corresponding disk time and downloading throughput. Figure 9(b) shows the the IO-busy percentage

and the downloading throughput when retrieving the files. As can be seen in this figure, the downloading throughput of the enhanced IPFS client outperforms the original IPFS is more resilient to the reduction in chunk size as a result of lower I/O overhead.

## 7 Related Work

**Decentralized Storage Network.** As a supplement to traditional cloud storage system, decentralized storage network (DSN) has been undergoing a fast development and widespread adoption as a result of the evolving blockchain technology. Besides IPFS, the most well-known DSNs are Storj [17], Sia [23], Filecoin [16] and Swarm [26], all of which aim to provide a censorship-resilient alternative to cloud storage. There are also prototypes of DSNs. For example, FileDAG [14] uses DAG-Rider as the consensus algorithm and builds a two-layer DAG-based blockchain ledger, facilitating flexible file indexing and multi-versioned file storage. As for the data management in DSNs, previous work have investigated load balancing [12, 25, 39], data security [13, 36, 38] and deduplication [35, 40, 41].

**IPFS.** As the most popular storage layer for Web3 applications, many prior studies have investigated various aspects of IPFS, ranging from measuring its performance [4, 20, 22, 27], analyzing its decentralization [6, 10, 22, 31], discussing its design and implementation [11, 15], exploring optimization methods [21, 24, 28, 37], and potential to support applications such as video streaming [32], among others. Previous works [6, 10, 22, 31] on the measurement of IPFS centralization are limited to snapshots of centralization at specific moments—essentially, the times at which the experiments are conducted. In contrast, our study aims to analyze the evolution of IPFS's centralization over a three-year period, presenting a more comprehensive view of centralization trends within IPFS rather than at isolated points.

## 8 Conclusion

IPFS showcases the Web 3.0 design and implementation. While many aspects of IPFS have been explored, a concerning trend reported by some prior studies shows that IPFS exhibits high degree of centralization when serving clients, contradicting its decentralization goal. In this paper, we have conducted a measurement and analysis study based on a 3-year trace collected from IPFS internal traffic. Our findings, including a low data replication level, a high and increasing degree of centralization, and negligible deduplication efficiency, offer a more complete view of the IPFS evolution over time, presenting both challenges and opportunities that IPFS (and Web 3.0) face(s). We hope our work can provide insights into the development and optimization of IPFS in the next stage.

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

## A   Ethics

This work is conducted under an IRB approval from our institution. Both the Bitswap and DHT traces contain IP addresses, yet our experiment does not attempt to map those IPs to any individuals or entities, as such analysis is not within the scope of our study. Furthermore, the IPs are anonymized and only mapped to contry-level for usage of geolocation mapping.

We also note that those Bitswap and DHT traces also contain personal browsing and publication history. However, we do not attempt to track any personal usage and collect any personal information. Although we download the CIDs for the purpose of deduplication analysis, we do not attempt to perform any content analysis on them and only seek to understand the deduplication level of its host node. All the downloaded CIDs are deleted immediately once the deduplication analysis is done. Furthermore, we recognize that CID downloading could potentially introduce additional load on the IPFS network. However, we argue that this impact is minimal. The downloading process is spread over a three-week period, generating an average of 55 GB daily traffic, which is negligible compared to the estimated over 100 TB daily traffic on the IPFS network [27].

