# OpenReview forum: "Centralization in Decentralized Web: Challenges and Opportunities in IPFS’s Data Management"
_ACM.org/TheWebConf/2025/Conference — WWW 2025 Poster_

### Official Review · Reviewer_erVR · 2024-11-10

**Novelty:** 6
**Technical Quality:** 5

**Review:**

Summary:
This paper presents a comprehensive measurement study of IPFS over a three-year period (2021-2024), analyzing over 20 billion messages. The study reveals three significant findings: (1) IPFS exhibits a low replication level with only 29.20% of content having more than one copy, (2) there is an increasing centralization trend where 5% of nodes host 80.55% of content, primarily driven by cloud nodes, and (3) the current deduplication strategy is inefficient. The authors propose a new metadata format to optimize deduplication while maintaining performance.

Strengths:
The paper addresses a crucial aspect of Web 3.0 by examining IPFS, a fundamental component of decentralized storage systems. The findings effectively demonstrate the gap between IPFS's design goals of decentralization and its actual implementation. The longitudinal study spanning three years provides valuable insights into the system's evolution, revealing concerning trends in centralization and inefficient data management. This work significantly contributes to our understanding of the practical challenges in implementing decentralized systems.

Weaknesses:
While the paper effectively identifies several critical issues in the IPFS system, it lacks comprehensive analysis and solutions for most of the discovered problems. Although it proposes a solution for the deduplication issue, it doesn't adequately address the more fundamental problems of low replication and increasing centralization. However, this limitation is somewhat understandable given the paper's primary focus on measurement and data mining rather than system design.

**Questions:**

1. Given that IPFS is a distributed system, to what extent can the collected data represent the entire IPFS ecosystem? Can this representativeness be quantitatively assessed?

**Reviewer Confidence:**

1: The reviewer's evaluation is an educated guess

**Scope:**

4: The work is relevant to the Web and to the track, and is of broad interest to the community

---

### Official Review · Reviewer_G7fd · 2024-11-29

**Novelty:** 5
**Technical Quality:** 5

**Review:**

Strengths:
1. The authors collected a substantial dataset spanning three years, comprising over 20 billion messages. This provides a robust foundation for analyzing long-term trends in IPFS’s centralization and data management.
2. The study not only investigates the centralization trend within IPFS but also delves into data replication levels and deduplication efficiency. This multi-faceted approach offers a holistic understanding of IPFS's data management dynamics.
3. The use of statistical measures such as the Gini Coefficient and Shannon Entropy to quantify centralization levels is methodologically sound and provides clear, quantitative insights into the distribution of data across the network.
4.  Proposing a new metadata format to optimize deduplication without compromising performance demonstrates practical applicability.

Weakness:
1.  The experiments were primarily conducted on specific EC2 instances, which may limit the generalizability of the results. Furthermore, the 4 vCPUs and 16GB memory configuration appears to be somewhat inadequate.
2. While the authors have designed a new metadata format, the accompanying evaluation is unduly particular and inadequate.

2. Please list questions for the authors for the discussion period, involving issues that an author response could change your opinion, clarify a confusion, or address a limitation.

**Questions:**

See the weaknesses part of the reviews. Furthermore, as I am only peripherally involved in the IPFS field and not an expert, I will try to refer to issues, confusions or limitations raised by other reviewers for further evaluation of this paper.

**Reviewer Confidence:**

3: The reviewer is confident but not certain that the evaluation is correct

**Scope:**

3: The work is somewhat relevant to the Web and to the track, and is of narrow interest to a sub-community

---

### Official Review · Reviewer_3JrF · 2024-11-29

**Novelty:** 3
**Technical Quality:** 3

**Review:**

The paper is based on three years of internal IPFS traffic data, analysing trends of centralisation, data replication levels, and deduplication efficiency, offering a well-rounded perspective. The study reveals that the degree of centralisation in IPFS has significantly increased, with 5% of nodes hosting 80% of the content, indicating a deviation from the decentralised goals of the system. To address the issue of low deduplication efficiency, the paper proposes a new metadata format that optimises the deduplication process while maintaining a balance with performance.

However, although centralisation can enhance performance, it may also introduce issues regarding data availability and fairness. The paper does not sufficiently explore these potential risks. And the data collection is limited to specific nodes and time periods, which may not fully represent the overall performance and behaviour of IPFS. Besides, the paper primarily focuses on technical analysis, ignoring a deeper discussion of user experience and practical application scenarios.

**Questions:**

What is your view on the impact of the current centralisation trend on the long-term sustainability and user trust in IPFS?

You mention that replication levels are relatively low in your study. What strategies do you think could effectively increase the data replication rate?

Has the new metadata format you proposed been tested in real-world applications? If so, what were the results?

**Reviewer Confidence:**

3: The reviewer is confident but not certain that the evaluation is correct

**Scope:**

3: The work is somewhat relevant to the Web and to the track, and is of narrow interest to a sub-community

---

### Official Review · Reviewer_EkDg · 2024-12-02

**Novelty:** 6
**Technical Quality:** 6

**Review:**

### Problem / Summary

- IPFS as a decentralized Web 3.0 suite of protocols has recently been shown to contradict its core ethos in its highly centralized nature. This work dives deep into IPFS usage over dimensions such as replication level, centralization trends, and deduplication methods to provide valuable context and help to improve IPFS moving forward
- The authors find low replication levels, growth in the centralization of IPFS, and massive room for improvement in deduplication

### Strengths

- Very well written
    - The layout of this paper is phenomenal, extremely easy to follow and read, with sections laid out in a great setup-details-takeaway format
    - Excellent background section, concise yet everything a reader needs
- Very information dense, yet extremely readable
- Provides new insights into IPFS usage
- No stone un-turned, each step in the paper makes sense to be making, and evaluations were thorough
- Already distilling findings into an actionable improvement on the deduplication process via new metadata structure

### Weaknesses

- Wish there was a bit more discussion surrounding how data collection methodology gives enough view to have the takeaways offered. (Section 3.1 - though the space constraints make that very difficult)
- The only place the 3 year dataset is utilized is in the centralization exploration
    - otherwise, analysis is still a snapshot
- Given the tight space constraints, I understand why, but I would have loved to get some insight as to the major players over time

### Nits

- Citations should at least include the conference, not just arXiv (for instance #6 was published at IMC ‘23)
- Page 2, RQ2: CID appears without prior definition
- Comma apocalypse

    > On the other hand, regardless of the underlying reasons, from a practical standpoint, such centralization, if it exists, could present an opportunity for improved web data management and cost savings through deduplication [33], which is important for those cloud nodes.
    >

**Questions:**

- Is the data in section 4 cumulative over the 3 year study period?
- In section 5

    > for those peers announcing multiple IP addresses as a result of peer churn, we count them as a single entity, i.e., provider.
    >
    - Could this choice bias your results towards more centralization? at least as compared to prior work?
- I’m curious at the rate of change in content, at what rate do you see new CIDs, is entropy affected not because of behavior but because of lack of new content?

**Reviewer Confidence:**

3: The reviewer is confident but not certain that the evaluation is correct

**Scope:**

3: The work is somewhat relevant to the Web and to the track, and is of narrow interest to a sub-community